# A Rapid Method for Postmortem Vitreous Chemistry—Deadside Analysis

**DOI:** 10.3390/biom12010032

**Published:** 2021-12-27

**Authors:** Brita Zilg, Kanar Alkass, Robert Kronstrand, Sören Berg, Henrik Druid

**Affiliations:** 1Forensic Research Laboratory, Department of Oncology-Pathology, Karolinska Institute, 171 77 Stockholm, Sweden; brita.zilg@ki.se (B.Z.); kanar.alkass@ki.se (K.A.); 2Department of Forensic Genetics and Forensic Toxicology, National Board of Forensic Medicine, 587 58 Linkoping, Sweden; robert.kronstrand@rmv.se; 3Division of Clinical Chemistry and Pharmacology, Department of Biomedical and Clinical Science, Faculty of Medicine and Health Science, Linköping University, 581 85 Linkoping, Sweden; soren.berg@liu.se

**Keywords:** vitreous, postmortem, glucose, electrolytes, forensic medicine

## Abstract

Vitreous fluid is commonly collected for toxicological analysis during forensic postmortem investigations. Vitreous fluid is also often analyzed for potassium, sodium, chloride and glucose for estimation of time since death, and for the evaluation of electrolyte imbalances and hyperglycemia, respectively. Obtaining such results in the early phase of a death investigation is desirable both in regard to assisting the police and in the decision-making prior to the autopsy. We analyzed vitreous fluid with blood gas instruments to evaluate/examine the possible impact of different sampling and pre-analytical treatment. We found that samples from the right and left eye, the center of the eye as well as whole vitreous samples gave similar results. We also found imprecision to be very low and that centrifugation and dilution were not necessary when analyzing vitreous samples with blood gas instruments. Similar results were obtained when analyzing the same samples with a regular multi-analysis instrument, but we found that such instruments could require dilution of samples with high viscosity, and that such dilution might impact measurement accuracy. In conclusion, using a blood gas instrument, the analysis of postmortem vitreous fluid for electrolytes and glucose without sample pretreatment produces rapid and reliable results.

## 1. Introduction

Autopsy has long been considered the gold standard in reaching a diagnosis when a person has died of uncertain causes. While this is still true regarding a large number of illnesses that are visible macroscopically or microscopically, there are many serious medical conditions that may escape detection. The major drawback is that autopsy diagnostics are traditionally based on morphology. Although computer tomography and magnetic resonance imaging has been introduced in the routine casework at many forensic medicine facilities in the last few decades, these radiological methods can also only provide structural information. Forensic toxicology is the only exception from the tradition of morphological diagnostics, which allows for the detection and quantification of alcohol and drugs by means of chemical analysis. With the help of postmortem reference concentrations, the pathologist may be able to diagnose, or rule out, an intoxication as the cause of death [1]. Toxicology was most likely introduced because intoxication does not usually cause any visible morphological signs/traces at autopsy.

Chemical analyses of postmortem samples today are not limited to toxicology; analysis of endogenous biomolecules in postmortem samples can also be used to identify pathologies. For instance, an increase in glial fibrillary acidic protein or neurofilament light protein in cerebrospinal fluid or serum can indicate brain injury [2] and increased troponin T [3] may be used as an indicator of myocardial infarction. Negative results can be equally important in the evaluation of the possible causes of death in the early stages of the investigation. Pioneering work in this field was performed by Coe [4,5] and over several decades, postmortem biochemistry has become increasingly appreciated as an asset in forensic pathology casework [6]. Most research in this field has explored the changes in various analytes in serum, pericardial fluid, cerebrospinal fluid, and vitreous fluid. The problem with postmortem serum is that it usually shows substantial hemolysis, which may interfere with, or even preclude, some analyses. In contrast, vitreous fluid is a transparent fluid with a very low amount of cells, at least during the early postmortem interval. It is also easily accessible before autopsy by a puncture at the lateral aspect of the eyeball. Hyperglycemia can be clearly identified by analyzing vitreous fluid glucose levels [7,8], whilst vitreous potassium levels can provide an estimate of the postmortem interval [9,10,11,12] and levels of sodium and chloride can reveal dehydration or water intoxication [13,14].

In hospitals, rapid analyses often form the basis for treatment measures in an emergency setting, whereas it is generally believed that urgent actions are not needed when a person dies. However, many biochemical changes start to occur during the agonal stages of death and the early postmortem interval. Glucose and oxygen are consumed until the cells die, and many cell functions, including membrane pumps which require energy, will fail as the ATP levels drop, at which stage an uncontrolled diffusion of ions and molecules flow in both directions. Therefore, postmortem biochemistry is easier to interpret if samples are collected as soon as possible after death. Early sampling and analysis are also important for the postmortem workup since results obtained in the early stages can form the basis for important decisions regarding how to proceed with the case examination and any additional sampling and analysis which may be warranted, just like in the emergency room. Even the police may want certain information urgently, such as an estimate of the time of death, and this can be reported quickly based on an analysis of vitreous potassium. We have used blood gas instruments to analyze postmortem vitreous fluid for 20 years and have reported our experience of using such analyses for the diagnosis of hyperglycemia, sodium and chloride imbalances and for the estimation of the postmortem interval by analysis of vitreous potassium [8,11,14]. There are also numerous other publications on these subjects, however the authors have usually sent samples to a clinical chemistry laboratory for analysis. The interpretation of vitreous chemistry results has been a matter of discussion; in particular, concerns have been raised regarding the possible influence of factors such as centrifugation, dilution and storage of the samples [15,16,17,18]. The possible impact of differences in concentrations between the eyes has also been an issue [17]. Even though it has been observed in forensic toxicology that the pre-analytical factors are much more important than the errors introduced during sample extraction and analysis [13]—which is most likely true for most biochemical analysis—it is important to understand the imprecision of point-of-care instruments, such as blood gas instruments. Hence, in this study, we aimed to investigate the reliability of blood gas instruments with regard to the results for potassium, sodium, chloride and glucose, in comparison with results obtained by analysis at external laboratories. We also studied the possible impact of sampling technique and the pretreatment/handling of the samples before analysis.

## 2. Materials and Methods

### 2.1. Sample Collection

Our standard procedure for collecting vitreous fluid is to puncture the lateral aspect of the eye and withdraw a 0.25 mL sample from the center of the vitreous compartment, using a 16-gauge needle into a 1 mL or 10 mL syringe. The tip of the needle is then typically visible through the pupil. For the current study, all of the vitreous fluid was collected, unless otherwise specified. The vitreous fluid is viscous, and becomes more liquefied with age [19]. However, even in younger subjects, it is possible to collect virtually all of the vitreous fluid if needed by using the procedure described above. For some of the analyses, the vitreous fluid was injected directly from the syringe by which the sample was collected. For test tube samples, the fluid was aspirated by the instrument.

### 2.2. Analytical Principles of Blood Gas Instruments

The blood gas instruments used for this study were ABL700, ABL835 and ABL90 flex (all from Radiometer Copenhagen, Brønshøj, Denmark). Each of these instruments is able to determine levels of Na^+^, Cl^−^, K^+^, Ca^++^, glucose, lactate, pH, pCO_2_, pO_2_, total Hb and specific forms of Hb (i.e., COHb, MetHb, O_2_Hb, HHb, HbF and SHb). For this project, the concentrations of potassium, sodium, chloride and glucose were measured. The analysis of these parameters was performed by sensor units with selective permeable membranes. The sensor units employ two different measuring principles, potentiometry for electrolytes and amperometry for glucose and lactate. For electrolyte measurements, the sensing element is an ion-selective sensor with a membrane with a specific ion carrier (potassium and chloride) or an ion-selective ceramic pin (sodium). The concentration of glucose was measured using an amperometric method. The sensor has a silver cathode and a platinum anode in contact with an electrolyte solution.

### 2.3. Precision Test of the Blood Gas Instrument

Prefabricated test solutions (Radiometer, Copenhagen) of glucose 4.4, 11.1 and 15.7 mmol/L and potassium 4.3 mmol/L were used, and additionally, solutions containing 10 and 30 mmol/L of potassium were prepared. The samples were measured using the ABL90 flex blood gas instrument (Radiometer Copenhagen). Samples of glucose and potassium with three different concentrations were aliquoted and analyzed eight times on one day (within-series imprecision) and one time on eight different days (between-series imprecision).

### 2.4. Comparison Different Instruments

Whole vitreous samples were collected from 50 consecutive postmortem cases. The samples were centrifuged and then divided into three aliquots. One sample was sent to an external lab (Aleris Medilab, Stockholm, Sweden) and was analyzed with a Beckman Coulter AU5800 instrument. There the samples were diluted 1:2 with water. Samples which had a potassium concentration exceeding the measuring range were diluted 1:4. Another aliquot was sent to a different external lab and analyzed with an inductively coupled plasma-mass spectrometry (ICP-MS) instrument at ALS Scandinavia, Luleå, Sweden (at the time of the study, it was named ALS Global). The third aliquot was analyzed undiluted with two blood gas instruments, ABL700 and ABL90 flex (Radiometer, Copenhagen).

### 2.5. Influence of Cells and Centrifugation

To investigate if cells present in whole vitreous samples affect the measurements of the electrolyte concentrations, 22 whole vitreous samples were studied. Each sample was vortexed for one minute. Then, 0.2 mL was aspirated and measured without further treatment using an ABL 835 blood gas instrument, and 0.015 mL was taken for cell counting. The remaining fluid was centrifuged for 10 min at 1900× *g*. The supernatant was collected until 0.2 mL of fluid remained in the tube. The pellet was vortexed again for one minute and then sonicated (Bransonic 12, Branson Ultrasonic Corporation, Danbury, CT, USA) for two hours. The samples from the supernatant and the pellet were also measured using the blood gas instrument. For the staining of cells, 0.015 mL of the sample was briefly vortexed with 0.015 mL of Trypan Blue (0.4%, Sigma Aldrich, Stockholm, Sweden, filtered and diluted 1:10). After 10 min of incubation, the fluid was filled into a Bürker chamber. The cells of several squares (0.1 mm^3^ each) were counted. The cell count was calculated using the following formula: counted cells × dilution factor × 10,000/number of C-squares = cells/mL. From this collection of samples, two groups of 6–8 samples with cell numbers >100,000 or <20,000 cells/mL were analyzed with ABL835.

### 2.6. Comparison of Undiluted and Diluted Samples

Whole vitreous fluid from 29 postmortem cases was sampled with a 16-gauge needle and a 1 mL syringe. A 0.2 mL portion was injected directly into the ABL700 and 0.2 mL was diluted with 0.2 mL of DDH_2_O water and vortexed. The samples were analyzed with an ABL700 blood gas instrument.

### 2.7. Impact of Hyaluronidase and Hyaluronic Acid

Vitreous fluid was collected from eight postmortem cases. Each sample was vortexed and 1 mL was aliquoted to each of three Eppendorf tubes. The content in one tube was untreated. To the second tube, 100 μL of distilled water was added and 100 μL of hyaluronidase was added to the third tube (Sigma, 50 mg/mL). All samples were well vortexed and incubated for 1 h at room temperature before analysis. To mimic the impact of a high concentration of hyaluronic acid, sodium hyaluronate (Hyalgan, Takeda Pharma, Stockholm, Sweden, 10 mg/mL) was added to duplicates of a prepared NaCl solution at increasing volumes.

### 2.8. Differences between Whole Vitreous and Sample from the Centre of the Vitreous

Vitreous samples from 27 postmortem cases were collected with a 16-gauge needle and a 10 mL syringe. The whole vitreous was withdrawn from the right eye. To this end, the needle usually had to be moved around to different positions within the vitreous compartment to enable complete sampling, and more than 2 mL could be obtained in most cases. Only 0.25 mL was aspirated from the left eye from the center of the vitreous; the tip of the needle was typically visible through the pupil. Both samples were measured immediately with an ABL 835 blood gas instrument using the 0.095 mL program. In addition to electrolytes and glucose, lactate concentrations were also registered.

### 2.9. Comparison of Samples Collected from the Left and Right Eye

In 16 postmortem cases, 0.25 mL of vitreous fluid was sampled from the center of each eye with a 16-gauge needle and a 1 mL syringe. Fluid was taken from the right and left eye in separate syringes and analyzed immediately with an ABL700 blood gas instrument.

### 2.10. Effect of Spiking of Vitreous Samples

In 33 postmortem cases, analysis was directly performed on an ABL700 blood gas instrument, and 0.3 mL of the remaining vitreous sample was transferred to an Eppendorf tube. A solution containing K, Na, Cl and glucose in the concentration of 70, 350, 420 and 160 mmol/L was made, and 30 μL of this solution was added to the aliquots of the vitreous samples. The results were compared with the calculated increase in concentrations of the solutes.

### 2.11. Effect of Delayed Analysis

Vitreous fluid from 13 cases was sampled with a 16-gauge needle and a 10 mL syringe. The samples were analyzed with an ABL90 blood gas instrument directly after sampling, without vortexing. The remaining portion of the samples were then stored in a refrigerator and analyzed again one or several days later, after being vortexed for 1 min. This test aimed to reflect the situation in practice; ideally the samples are analyzed immediately after collection, but occasionally, the analysis cannot be done until a few days later for various reasons.

### 2.12. Statistical Methods

The Mann–Whitney U-test was used for comparison between groups. ANOVA was used for comparisons between multiple groups. The association between variables was analyzed with linear regression. A *p* < 0.05 was considered statistically significant. All statistical analyses were performed with SPSS v. 25 (SPSS Inc., Chicago, IL, USA).

## 3. Results

### 3.1. Imprecision of the Blood Gas Instrument

Table 1 shows the within-series and between-series imprecision. The CV% for the within-series imprecision was very low, and was maximally 3.24% between-series for the analysis of 10.0 mmol/L potassium. Please note that the calibration interval of the instrument was 1–25 mmol/L for potassium, and high precision was still obtained for the 30 mmol/L solution. For potassium, almost all within-series measurements showed the same concentration for each test solution.

### 3.2. Comparison between Different Analytical Instruments

In Figure 1, Figure 2, Figure 3 and Figure 4 the results obtained with the ABL90 flex and other instruments are shown. Almost identical results were obtained with the ABL 90 flex and ABL700 (Figure 1). Analysis with a Beckman Coulter AU5800 showed significantly (*p* < 0.001) higher sodium concentrations (Figure 2), but very similar potassium and chloride concentrations. ICP-MS analysis showed consistently lower sodium concentrations compared with the ABL 700 (*p* = 0.002), ABL90 flex (*p* < 0.001) and Beckman Coulter AU5800 (*p* < 0.001) and also higher chloride concentrations than the ABL90 flex (*p* < 0.019) and Beckman Coulter AU5800 (*p* = 0.007) (Figure 3 and Figure 4). Up to about 20 mmol/L, the ICP-MS showed similar potassium concentrations as the ABL 90 flex and Beckman Coulter AU5800, but showed an upward deviation at higher potassium levels.

### 3.3. Influence of Cells/Centrifugation

Two groups of samples with >100,000 and <20,000 cells/mL were analyzed; the median was 1,097,500 and 5000 cells/mL for the group with a high and low cell-count number, respectively. Table 2 shows the means and the standard deviations of the electrolytes for each group of samples. There were no significant differences in electrolyte concentrations between untreated samples, and supernatant and pellets of the centrifuged samples in either group (Mann–Whitney U-test). The higher levels of potassium and lower levels of sodium and chloride in the group with the high cell number reflect the longer postmortem interval in this group, which is why more desquamated cells were found.

### 3.4. Comparison of Undiluted and Diluted Samples

Figure 5 shows undiluted vitreous samples compared with samples that were diluted 1:2 with distilled water. Dilution seems to have no impact on potassium or glucose, whereas sodium and chloride show slightly lower values than expected when diluted. Please note that the diluted concentrations for sodium and chloride are below the calibration range of the ABL700 instrument.

### 3.5. Hyaluronan, Hyaluronidas

Almost identical results were obtained for potassium, sodium and chloride when adding hyaluronidase or distilled water (Table 3a). When corrected for the dilution factor, these results were very similar to the analyzed results of the untreated sample. The addition of sodium hyaluronate to water solutions of these electrolytes caused a gradual increase in the concentrations of sodium and chloride, which was proportional to the content in the added hyaluronate (dissolved in 145 mmol/L NaCl) (Table 3b), whereas potassium showed the same values as the samples with the addition of distilled water.

### 3.6. Differences between Whole Vitreous Fluid and Vitreous from the Center of the Eye

The analytical results based on the 0.25 mL sample collected from the left eye and the whole vitreous sample collected from the right eye are shown in Figure 6. There was a close correlation (r^2^ = 0.97–1.00, slope 1.03–1.09) for potassium, chloride and glucose, whereas sodium levels were somewhat more dispersed (r^2^ = 0.86, slope 0.86). No significant differences between the central sample and the whole vitreous sample were seen with regards to electrolyte concentrations, glucose or lactate (ANOVA and Mann–Whitney U-test, *p* > 0.05). Since it is known that lactate increases in vitreous fluid with PMI [8,20], and that postmortem diffusion might cause an uneven distribution, we included lactate levels in this comparison but found no difference in concentrations between the central sample and whole vitreous sample.

### 3.7. Comparison of Samples Taken from the Left and Right Eye

The results of whole vitreous samples collected from the right and left eye correlated well (Figure 7). Linear regression showed r^2^ of 0.95–0.99 with a slope coefficient of 0.93–1.02. There were no significant differences in the concentration of any of the solutes between the two groups of samples (Mann–Whitney U-test, *p* > 0.05).

### 3.8. Spike Test

In Figure 8, the effects of spiking postmortem vitreous samples with electrolytes and glucose are shown. At higher concentrations, the measured sodium and chloride concentrations showed a trend towards higher values than the calculated concentration. However, as can be seen in the Bland–Altman plots, there was only little overall bias between the measured and calculated values for each of the analytes.

### 3.9. Comparison of Vortexed and Unvortexed Samples

Figure 9 shows the results for vitreous samples that were analyzed immediately after sampling, unvortexed, and for samples that were analyzed one or several days later after being vortexed. Neither the delay in analysis nor vortexing had any effect on the measurements of glucose or any of the electrolytes.

## 4. Discussion

Given that it is well understood that a postmortem investigation benefits from relevant medical information in the early stages in supporting decision-making during, or even before an autopsy, it is remarkable that rapid biochemical analysis is not applied more often in routine forensic casework. Such “deadside” analysis can be regarded as being as important as bedside analysis in the clinical setting. Already in 1966, Reh reported a rapid method for the detection of diabetic coma by analyzing glucose in cerebrospinal fluid with an enzymatic method [21]. Zugibe (1966) similarly analyzed sodium and potassium in samples from cases of acute myocardial infarction using flame photometry and found that the sodium/potassium ratio was increased [22]. Despite such possible use of in-house, simple analysis of electrolytes and glucose, forensic medical facilities tend to rely on the services of local clinical chemistry laboratories. This is most likely based on the assumption that results from such laboratories are more reliable, given that their methodology is used extensively to analyze samples from living patients.

In this study, we investigated the feasibility of analyzing postmortem vitreous fluid with a blood gas instrument and we conducted several tests and comparisons to validate this strategy as a proof of concept. We showed that the imprecision of the ABL 90 flex instrument was very low for glucose and potassium (Table 1). Indeed, the CV% for potassium was no higher than 3.24, and this was for only one of the concentrations in the between-series study. Vitreous potassium is used extensively for estimating the postmortem interval (PMI), particularly when the true interval is so long that the classical methods used in the early phase no longer work. If we assume the measuring error to be 4%, i.e., a value of 12.5 mmol/L is obtained rather than 12.0, then the estimated PMI using the equations of Zilg et al. [11], Madea et al. [23], Sturner and Gantner [24] and Bortolotti et al. [25] would be 40.9, 34.8, 50.2 and 58.8 h, respectively, instead of 37.8, 32.2, 46.6 and 55.9 h, respectively. This example shows that the difference in results of these and many other equations surpass the analytical error. At very long postmortem intervals, the effect becomes even more pronounced.

We also showed that two blood gas instruments, the ABL90 flex (a simple instrument with cassette system) and a regular instrument, the ABL700, produced indistinguishable results in a large set of postmortem vitreous samples (Figure 1). When comparing the results of the ABL90 flex and Beckman Coulter AU5800, the latter instrument reported higher sodium values, but similar potassium and chloride results. The ICP-MS instrument used by ALS Global also provided similar potassium values, but generally lower potassium and higher chloride values. ICP-MS and atomic absorption spectrophotometry is supposed to provide the “true” numbers of atoms [26], but this is not necessarily what we want to know. Rather, we want to know whether a test result can be compared with established reference levels for a certain method. If there is an understanding of what the deviations mean, such a method should be feasible for clinical practice as well as in postmortem casework. By and large, both the blood gas instruments and the Beckman Coulter provided similar results, and we have previously shown the feasibility of the blood gas instrument readings of electrolytes and glucose in postmortem casework [8,11,14]. The difference between the ABL90 flex and Beckman Coulter AU5800 regarding sodium was statistically significant, but since the difference is not large, neither numerically or in terms of percentage, the interpretation of dehydration (or overhydration) will not be affected.

At longer PMIs, the vitreous fluid volume is reduced due to evaporation, and the fluid gradually shows an increasingly grey-brown tinge due to increased amounts of cells and cell debris, prompting other researchers in this field to centrifuge the samples to obtain a more transparent fluid for analysis. Thus, the question is if the cells present in the fluid retain a higher content of potassium, and lower content of sodium and chloride than the extracellular fluid. We therefore analyzed untreated vitreous fluid samples, and compared these with the supernatant and pellet after centrifugation. We found that there was no difference in concentrations in these three aliquots (Table 2), implying that the ion concentrations had equilibrated before the cells had detached.

Whenever high concentrations are found, and exceed the calibration range for a method, the standard procedure is to dilute the sample appropriately to obtain a concentration that matches the measuring range. Hence, we investigated the effect of a relevant dilution of the samples and found that the results were not much affected (Figure 5). While the potassium levels were not affected, there was a reduction in both the sodium and chloride levels. The most likely explanation for this is that dilution causes a reduction in the concentration levels, which are below the calibration range for sodium and chloride, but within the calibration range for potassium.

Concerns have been raised as to the possible impact of the viscosity of vitreous fluid on the accuracy of the analytical results. Blana et al. [16] reported that heat and hyaluronidase treatment of vitreous samples resulted in a slight increase and decrease, respectively, of measured electrolyte concentrations. We also investigated the possible impact of hyaluronidase treatment on electrolyte results, and found no differences compared to the results of untreated samples. Further, we studied the possible effect of the addition of hyaluronidase to water solutions of sodium and chloride, but found no influence on the analytical results, even at hyaluronidase concentrations that exceed those that are present in the vitreous fluid of phakic eyes [27].

Regarding the collection of vitreous fluid, we collected a small amount, approximately 0.25 mL, from each eye, which was pooled in the same syringe in order to still retain sufficient fluid for toxicological analysis. However, most other investigators have used whole vitreous samples for analysis. Given that the diffusion of ions and other solutes from the retinal cells after death may produce different concentrations in the central and peripheral compartments of the vitreous, we compared the results of analysis of samples collected from the central vitreous with the results of analysis of the whole vitreous. Figure 7 shows that almost the same concentrations of electrolytes was found in both samples.

Furthermore, we tested whether spiking authentical postmortem samples with electrolytes and glucose would produce results that correlated with mathematical calculations. Figure 8 shows that the results did not deviate much, implying that the matrix effects of hyaluronic acid and other molecules prevalent in the vitreous do not have any substantial impact on the distribution of either electrolytes or glucose.

Finally, we performed a test that should resemble the situation in practice, i.e., the possible impact of delay of analysis and sedimentation of particles in the solution. Samples that were stored in a refrigerator for one day were vortexed before analysis. We did not find any change in the concentrations of the analytes using such a practice (Figure 9). This means that though it is desirable to obtain a result as soon as possible, similar results will be obtained if the analysis is performed a day after sampling.

Today, most instruments use ion-selective electrodes for determining the concentrations of potassium, sodium and chloride. It is therefore not surprising that we observed similar results with blood gas instruments and a multi-analysis instrument that is typically used in clinical chemistry laboratories. The small difference in concentrations and any imprecision is most likely due to different lining conditions, implying that samples with higher viscosity had to be diluted to avoid clogging, and hence the concentrations measured fell short of the optimal measuring range of the analytes. Given such concerns regarding the internal conditions of laboratory instruments, it is tempting to consider simple hand-held instruments, such as an i-STAT^®^ analyzer. Monzon et al. (2018) evaluated the feasibility of such an analyzer in regard to electrolytes and glucose in vitreous and found low imprecision. However, the calibration range for potassium was reported to be 2.0–9.0 mmol, implying that many samples would have to be diluted before analysis [28].

## 5. Conclusions

We have shown that analysis of electrolytes and glucose with blood gas instruments in postmortem samples, provides rapid and reliable results without dilution, enzymatic treatment or centrifugation. We also found no difference in the concentrations of the electrolytes or glucose in samples from the right and left eye, and the results in samples from the central part of the vitreous were the same compared to the results of whole vitreous samples. We therefore conclude that a small, pooled sample from the center of each eye is feasible for these chemical analyses, which leaves a sufficient amount of vitreous fluid for toxicological analysis.

## Figures and Tables

**Figure 1 biomolecules-12-00032-f001:**
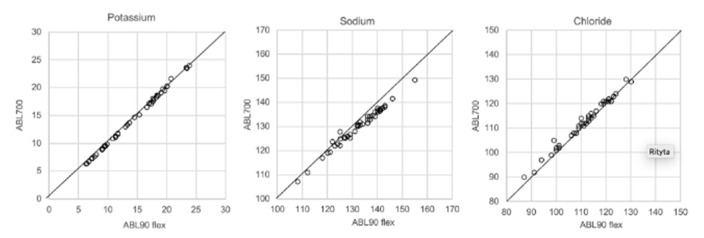
Comparison results using ABL90 flex and ABL700.

**Figure 2 biomolecules-12-00032-f002:**
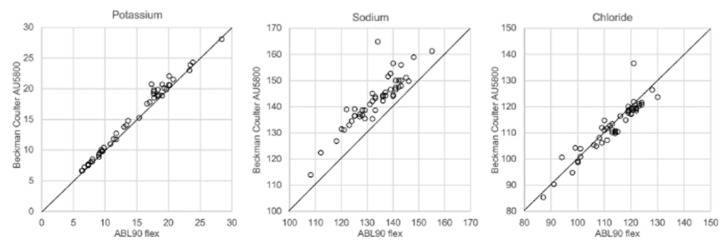
Comparison of ABL90 flex and Beckman Coulter AU5800.

**Figure 3 biomolecules-12-00032-f003:**
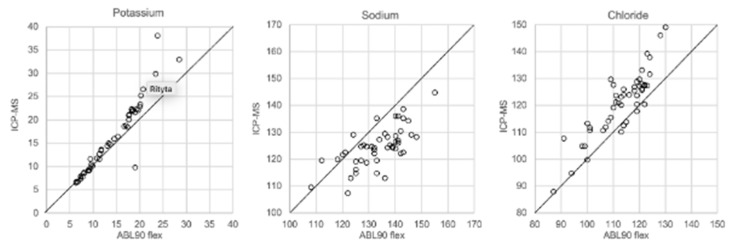
Comparison of ABL90 flex and ICP-MS.

**Figure 4 biomolecules-12-00032-f004:**
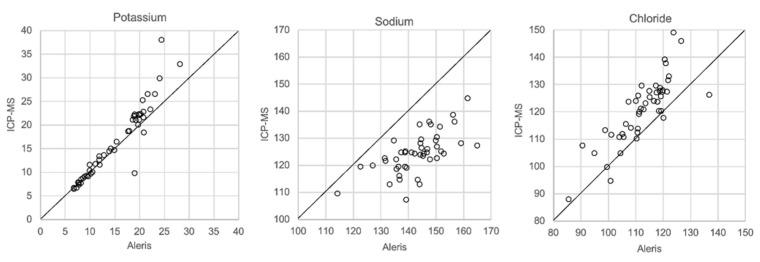
Comparison of Beckman Coulter AU5800 and ICP-MS.

**Figure 5 biomolecules-12-00032-f005:**
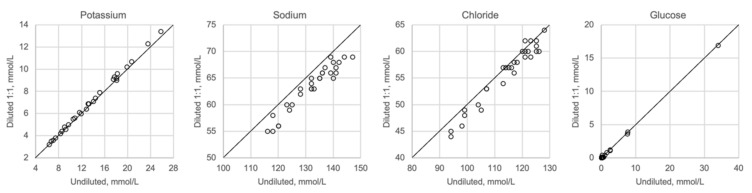
Comparison of undiluted and diluted samples, *n* = 29.

**Figure 6 biomolecules-12-00032-f006:**
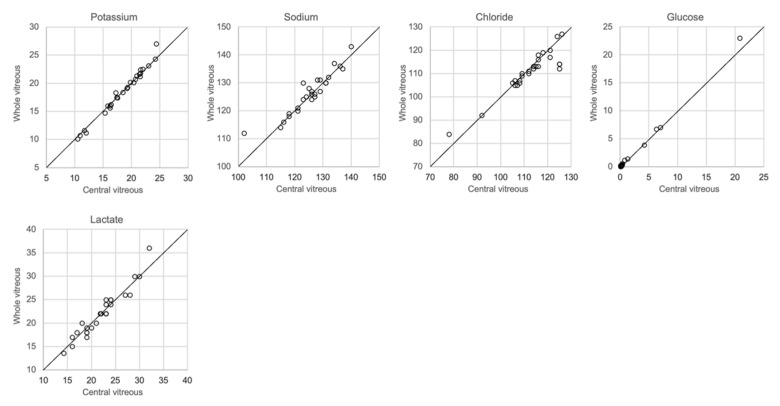
Comparison of whole and central vitreous.

**Figure 7 biomolecules-12-00032-f007:**
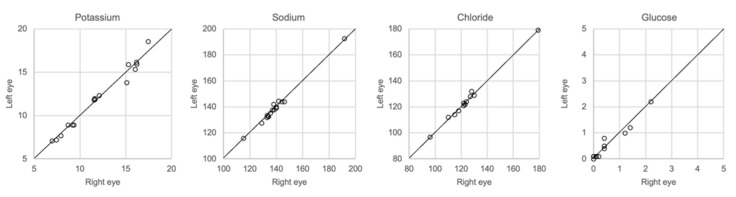
Comparison of samples taken from center of the right and left eye, *n* = 16.

**Figure 8 biomolecules-12-00032-f008:**
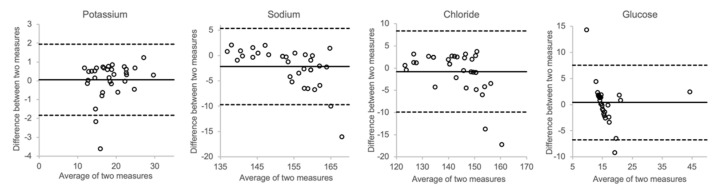
Comparison of actual concentration and calculated concentration (vitreous + spike).

**Figure 9 biomolecules-12-00032-f009:**
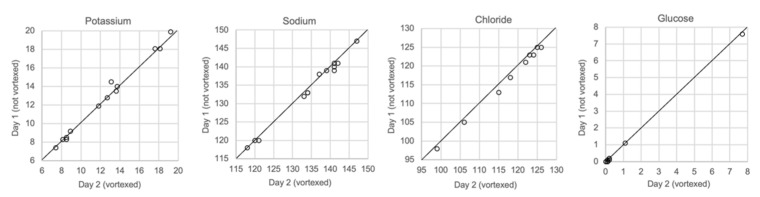
Comparison of samples not vortexed (day 1) and vortexed (day 2), *n* = 13.

**Table 1 biomolecules-12-00032-t001:** Within-series and between-series imprecision of ABL90 flex instrument.

Within-Series Imprecision							Between-Series Imprecision						
	Glucose			Potassium				Glucose			Potassium		
	Solution 1 4.4 mmol/L	Solution 2 11.1 mmol/L	Solution 3 15.7 mmol/L	Solution 2 4.3 mmol/L	10 mmol/L	30 mmol/L		Solution 1 4.4 mmol/L	Solution 2 11.1 mmol/L	Solution 3 15.7 mmol/L	Solution 2 4.3 mmol/L	10 mmol/L	30 mmol/L
Mean	4.26	10.55	15.13	4.40	10.00	29.96	Mean	4.26	10.61	15.23	4.41	9.75	29.49
SD	0.07	0.09	0.15	0.00	0.00	0.07	SD	0.07	0.23	0.31	0.04	0.32	0.55
CV%	1.75	0.88	0.98	0.00	0.00	0.25	CV%	1.75	2.16	2.01	0.80	3.24	1.87
Bias%	−3.85	−4.81	−3.86	2.33	0.00	−0.12	Bias%	−3.85	−4.25	−3.23	2.62	−2.50	−1.71

**Table 2 biomolecules-12-00032-t002:** Impact of cells and centrifugation. Means and standard deviations of the concentrations of potassium, sodium and chloride.

	Cell Number (Cells/mL)	Untreated (mmol/L)	Supernatant (mmol/L)	Pellet (mmol/L)
K^+^	>100,000 (*n* = 6)	19.8 ± 3.8	19.9 ± 3.7	19.8 ± 3.4
	<20,000 (*n* = 7)	16.9 ± 3.5	17.0 ± 3.5	17.2 ± 3.6
Na^+^	>100,000 (*n* = 8)	120.8 ± 13.1	122 ± 12.9	123 ± 12.7
	<20,000 (*n* = 7)	129.1 ± 8.7	130 ± 9.0	132 ± 9.1
Cl^−^	>100,000 (*n* = 8)	104 ± 10.9	104 ± 11.2	109 ± 13.2
	<20,000 (*n* = 7)	111 ± 10.8	111 ± 11.1	114 ± 11.1

**Table 3 biomolecules-12-00032-t003:** Effect of addition of (**a**) hyaluronidase to postmortem vitreous samples and (**b**) sodium hyaluronate to water solutions of electrolytes.

(**a**)													
**Case #**	**Sample + DDH_2_O**			**Concentration Ratios**			**Case #**	**Sample + Hyaluronidase**			**Concentration Ratios**		
	**K^+^**	**Na^+^**	**Cl^−^**	**K^+^ Ratio**	**Na^+^ Ratio**	**Cl^−^ Ratio**		**K^+^**	**Na^+^**	**Cl^−^**	**K^+^ Ratio**	**Na^+^ Ratio**	**Cl^−^ Ratio**
1	11	130	107	0.92	0.92	0.93	1	10.9	129	106	0.92	0.91	0.92
2	11.6	130	96	0.94	0.93	0.92	2	11.6	129	96	0.94	0.92	0.92
3	7.8	127	110	0.93	0.92	0.92	3	7.8	127	110	0.93	0.92	0.92
4	13	122	107	0.92	0.91	0.91	4	13.1	122	107	0.93	0.91	0.91
5	10.6	119	106	0.92	0.92	0.93	5	10.5	118	105	0.91	0.91	0.92
6	11.9	126	111	0.92	0.91	0.92	6	11.9	126	111	0.92	0.91	0.92
7	16.3	122	105	0.93	0.92	0.92	7	16.3	122	106	0.93	0.92	0.93
8	21.1	95	77	0.92	0.91	0.92	8	21	95	77	0.92	0.91	0.92
(**b**)													
**1 mL solution +**				**K^+^**		**Na^+^**		**Cl^−^**		**Na^+^ Corr**		**Cl^−^ Corr**	
10 µL DDH_2_O				9.7		110		113		111		114	
10 µL DDH_2_O				9.7		110		113		111		114	
20 µL DDH_2_O				9.6		109		112		111		114	
20 µL DDH_2_O				9.6		108		112		110		114	
50 µL DDH_2_O				9.4		105		109		110		114	
50 µL DDH_2_O				9.4		105		109		110		114	
100 µL DDH_2_O				9.0		101		104		111		114	
100 µL DDH_2_O				8.9		100		104		110		114	
10 µL hyaluronate				9.7		111		115		112		116	
10 µL hyaluronate				9.7		111		115		112		116	
20 µL hyaluronate				9.7		112		115		114		117	
20 µL hyaluronate				9.7		112		115		114		117	
50 µL hyaluronate				9.4		113		116		119		122	
50 µL hyaluronate				9.4		113		116		119		122	
100 µL hyaluronate				9.1		115		117		127		129	
100 µL hyaluronate				9.0		115		117		127		129	

## Data Availability

All relevant data are provided in the article.

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
