# Peer review of "A Rapid Method for Postmortem Vitreous Chemistry—Deadside Analysis"

_biomolecules, 2021, doi:10.3390/biom12010032_

Round 1

Reviewer 1 Report

Biomolecules -1513458-peer-review-v1

A rapid method for postmortem vitreous chemistry – deadside analysis.

The article describes the analysis of vitreous fluid for the evaluation of electrolytes and metabolites such as potassium, sodium, chloride and glucose and lactate. The main aspects of the research were: sample characteristics and collection procedure comparing among others right and left eye, whole and middle vitreous and sample stability, dilution, etc. ; the effects of adding hyaluronidate and its catabolic enzyme hyaluronidase were studied. Analytical methods and instruments were also studied. The present research is certainly interesting and well conducted. What is lacking is a greater focus on the analytes studied, also considering the scientific literature. It is worthy of serious consideration and accepted for publication in the journal. Some minor revisions need to be made.

I suggest a variation of the title, tentatively: A rapid method for chemical analysis of vitreous for postmortem forensic pathology – deadside analysis.

Abstract

The abstract requires more details, in particular: a suitable description of materials and methods is required; the obtained results are too general (which biochemical parameters are studied and described? What results related to these parameters? namely potassium, sodium chloride and glucose). Moreover, a more “precise/accurate” conclusion would also be required.

Introduction:

The research compares different instruments and the influence of centrifugation, dilution and describes the possible effects of biofluid collection, storage, and treatment. These aspects are not considered and mentioned in the introduction section. In addition, particularly recent literature is not as widely cited (i.e., Belsey SL, 2016; Cordeiro C, 2019; Pagaiani N, 2020) regarding the general and more specifically forensic aspect of the vitreous as a biological fluid suitable for forensic purposes. The PMI just introduced in the abstract is not mentioned at all in the introduction (Rognum TO, 2016). Please, add critical presentation methods and existing data in the literature of each studied analyte. 

Materials and methods

This section is well described. I would suggest a specific paragraph devoted to the number and type of subjects from whom specimens are collected and the pathology present and causes of death (if it is deemed to interfere with the purpose of the investigations). In addition, the previously mentioned post-mortem interval (PMI) data are required as they are discussed in the manuscript (Hostiuc S at al 2021; Bertaso A et al, 2020). Please, add the proposed rapid method in more detail.

Results:

I am not a statistician but given the quality and large amount of data I would suggest an extension of the statistical analysis for more evidence of their validity.

Discussion:

The discussion is certainly valid and would be improved by a more detailed statistical analysis.

Please clarify what "normal" means, perhaps it should be explained if it refers to a range of measurements made in vivo or just referring to the analytical procedure. Or something else?

Author Response

Please see our response in the attachment.

Reviewer 2 Report

The authors show that vitreous analysis with blood gas instruments for potassium, sodium, chloride, and glucose quantification in postmortem samples provides rapid and reliable results.

The main question addressed by the research is about the analysis performance of vitreous fluid with blood gas instruments. In other papers with other instrumentations, the researchers described the electrolytes analysis as less sensitive, time-consuming, and complex.

The analysis simplicity and short-time results are the topics. The researchers collected the vitreous to the test tubes where the instrument aspirated the fluid.

The paper is well written, and the text is clear and easy to read. In other words, it clearly describes the analytical purposes, the precision intra labs and inter labs.

The conclusions are consistent with the evidence, and the arguments presented.

Suggestions

Detection and quantification of electrolytes in the vitreous is a specific tool determining the Post Mortal Interval (PMI).  The authors could have been studied the same samples at different intervals in different conservation conditions to observe the electrolytes concentration variation.

In line 349, after the sentence "The longer PMI", there is a colon ":", the authors should erase the colon.

Author Response

(The authors gave the same response as above.)
